# Evaluation of a Live Attenuated *S. sonnei* Vaccine Strain in the Human Enteroid Model

**DOI:** 10.3390/pathogens10091079

**Published:** 2021-08-25

**Authors:** Giulia Pilla, Tao Wu, Christen Grassel, Jonathan Moon, Jennifer Foulke-Abel, Christoph M. Tang, Eileen M. Barry

**Affiliations:** 1Sir William Dunn School of Pathology, University of Oxford, South Parks Road, Oxford OX1 3RE, UK; giulia.pilla@path.ox.ac.uk (G.P.); christoph.tang@path.ox.ac.uk (C.M.T.); 2Center for Vaccine Development and Global Health, University of Maryland School of Medicine, 685 West Baltimore Street, Baltimore, MD 21201-1509, USA; twu@som.umaryland.edu (T.W.); cgrassel@som.umaryland.edu (C.G.); jmoon@som.umaryland.edu (J.M.); 3Department of Medicine, Division of Gastroenterology and Hepatology, Johns Hopkins University School of Medicine, Baltimore, MD 21201, USA; jfoulke@jhmi.edu

**Keywords:** *Shigella sonnei*, human enteroids, *Shigella* vaccine

## Abstract

*Shigella* is a leading cause of bacillary dysentery worldwide, responsible for high death rates especially among children under five in low–middle income countries. *Shigella sonnei* prevails in high-income countries and is becoming prevalent in industrializing countries, where multi-drug resistant strains have emerged, as a significant public health concern. One strategy to combat drug resistance in *S. sonnei* is the development of effective vaccines. There is no licensed vaccine against *Shigella,* and development has been hindered by the lack of an effective small-animal model. In this work, we used human enteroids, for the first time, as a model system to evaluate a plasmid-stabilized S. *sonnei* live attenuated vaccine strain, CVD 1233-SP, and a multivalent derivative, CVD 1233-SP::CS2-CS3, which expresses antigens from enterotoxigenic *Escherichia coli*. The strains were also tested for immunogenicity and protective capacity in the guinea pig model, demonstrating their ability to elicit serum and mucosal antibody responses as well as protection against challenge with wild-type *S. sonnei*. These promising results highlight the utility of enteroids as an innovative preclinical model to evaluate *Shigella* vaccine candidates, constituting a significant advance for the development of preventative strategies against this important human pathogen.

## 1. Introduction

*Shigella* is responsible for a substantial burden of disease in multiple populations within the U.S. and worldwide, causing an estimated 163 million cases and more than 74,000 deaths per year [1,2]. The greatest impact is in children under 5 years of age in low- and middle-income countries, where *Shigella* was found to be the number one pathogen causing moderate to severe (MSD) diarrhea in children between 12–59 months of age and the fourth most important MSD pathogen in the youngest (0–11 months) children [3,4]. More recently, *Shigella* has also been recognized as a cause of considerable disease in the U.S. and other high-resource settings [5]. The widespread isolation of multi-drug resistant (MDR) isolates that limit therapeutic interventions and the continued high levels of endemic disease make *Shigella* a high priority for new preventative interventions and vaccine development.

*S. flexneri* is the most important strain causing disease in endemic regions, while *S. sonnei* causes the most disease in high-resource settings [6]. *S. sonnei* is gaining importance as regions transition to increased development, and *S. sonnei* replaces *S. flexneri* as the predominant pathogen [7,8,9]. MDR *S. sonnei* strains, refractory to most antibiotics, circulating at high levels in particular populations, is becoming recognized as a serious concern [10,11,12]. Within *S. flexneri,* there are more than 15 serotypes defined by the O-antigen structure (O-Ag) of LPS, while *S. sonnei* only has a single serotype [13]. Protective immunity is believed to be O-Ag serotype specific, and vaccine strategies (live attenuated or O-Ag-based conjugates) must account for the diversity of serotypes required to confer broad protection; a successful vaccine must cover *S. flexneri* and *S. sonnei* [12,14,15].

There are multiple strategies currently being pursued for *Shigella* vaccine development [12,14,15,16]. One approach that has shown promise in field trials and volunteer studies is live attenuated vaccines that can be delivered by the oral route. We have engineered a live attenuated strain of *S. sonnei*, CVD 1233, that contains a deletion in the *guaBA* operon, which encodes critical enzymes in the biosynthetic pathway of guanine nucleotides, and in the *sen* gene encoding *Shigella* enterotoxin 2 (ShET2). These deletions render *S. sonnei* auxotrophic for guanine and attenuated in vitro and in vivo [17]. One characteristic of *S. sonnei* strains is the frequent spontaneous loss of the large virulence plasmid, pINV, which is required for vaccine strain efficacy. To prevent this, we introduced mutations in one of the plasmid-encoded maintenance systems, which rendered pINV more stable [18]. The stabilized strain, CVD 1233-SP, was also further modified to express two heterologous antigens from enterotoxigenic *Eschericia coli* (ETEC), CS2 and CS3, as a component of a multivalent *Shigella*-ETEC vaccine, CVD 1233-SP::CS2-CS3.

The lack of preclinical models that reflect relevant human disease pathophysiology of *Shigella* have delayed the development of new vaccines and therapeutics. The human enteroid/colonoid “mini-gut” model presents a technological leap in human gastrointestinal (GI) system modelling [19,20,21]. Isolated human adult intestinal epithelial stem cells obtained from biopsies or surgically resected tissue can be indefinitely propagated as small intestinal epithelial cultures termed enteroids. Protocols have been established to grow human enteroid monolayer cultures on transwell filters that allow controlled access to both apical and basolateral surfaces. Differentiation leads to the increased expression of goblet and enteroendocrine cells as well as mature, nutrient-absorptive enterocytes. We optimized the enteroid model for the study of *Shigella* pathogenesis and showed preferential basolateral versus apical invasion, intracellular replication, and induction of host responses including IL-8 and the mucin glycoprotein MUC2 [21,22]. In this report, we describe the development and evaluation of a pINV-stabilized S. *sonnei* vaccine strain CVD 1233-SP and a multivalent derivative CVD 1233-SP::CS2-CS3 in the human enteroid model.

## 2. Results

### 2.1. Increased Stabilization of pINV Improves the Consistency of S. sonnei Vaccine Strain Production

The attenuated *S. sonnei* vaccine candidate strain CVD 1233 was constructed by introducing deletion mutations in the *guaBA* operon that render the bacteria auxotrophic for guanine and unable to replicate in the absence of this amino acid as well as in the *sen* gene, which prevents production of *Shigella* enterotoxin 2 (ShET2). These mutations result in attenuation of virulence [17]. Invasion and intracellular replication of this vaccine strain was tested in the human intestinal epithelial cell line HT29 and compared with the wild-type parent strain *S. sonnei* 53G (53G WT) (Figure 1A,B). As *S. sonnei* is known to spontaneously lose its 210 kb (pINV) at high rates due to instability [18], strains were cultured on agar containing Congo Red (CR agar) prior to infection to detect maintenance of pINV. Only colonies harbouring an intact pINV bind the CR and appear red (here defined as pINV+); in contrast, avirulent, plasmid-less colonies do not bind CR and appear white and rough (here defined as pINV-) due to the loss of pINV-encoded T3SS and O-Ag synthesis genes, respectively [18,23]. To minimise the impact of the loss of virulence, only pINV+ colonies were selected and used for inoculating HT29 epithelial cells. HT29 cells were incubated with bacteria for 90 min, then treated with gentamicin to remove extracellular bacteria and incubated for an additional 30 min or 4 h. Total intracellular bacteria were quantified by counting colony-forming units (CFU) in cell lysates at each time point (Figure 1A), and the fold increase of bacterial recovery at 4 h vs. 30 min incubation was also calculated for each strain (Figure 1B). To assess pINV stability of the strains, CFU were recovered from cells on media containing CR, and pINV+ and pINV- CFU were quantified (Figure 1C).

While at 30-min post infection, a higher recovery of CVD 1233 compared to 53G WT was evident, at 4-h post infection, a significant increase in recovery of intracellular bacteria was only observed with 53G WT but not CVD 1233 (Figure 1A). In fact, 53G WT was recovered at a 14-fold higher level at 4 h compared with 30 min, while CVD 1233 showed only a 1.6-fold increase. Due to high variation between experiments, the two strains did not show statistical difference in fold increase of bacterial recovery for intracellular replication, although CVD 1233 demonstrated a reduced increase in bacterial recovery compared to 53G WT, suggesting that this strain is unable to replicate intracellularly (Figure 1B).

We hypothesized that the high variation in bacterial invasion after 30 min of incubation with either strain could be due to the loss of pINV during growth in vitro. In fact, colonies selected from overnight cultures and used for infecting epithelial cells may contain a small proportion of pINV- bacteria that, when incubated with HT29 cells, would be incapable of invading cells, leading to a reduction in the total number of intracellular bacteria. The abundance of pINV- bacteria may vary from colony to colony and would not be quantifiable due to limitations of the CR binding selection, causing the aforementioned variation in bacterial invasion. Therefore, we quantified pINV- vs. pINV+ bacteria of CVD 1233 and 53G WT to assess the levels of spontaneous loss of pINV. We found that at 30-min post infection, more than 25% of bacteria lost pINV in CVD 1233, while more than 90% of 53G WT bacteria were pINV- (Figure 1C). At 4-h post infection, a lower proportion of pINV- bacteria was recovered, but pINV- bacteria still represented approximately 6% and 4.5% of the population of 53G WT and CVD 1233, respectively (Figure 1C).

To increase the stability of pINV in CVD 1233, we introduced mutations in one of the plasmid-encoded maintenance systems to render pINV more stable, creating CVD 1233-SP (Stabilized Plasmid) [18]. The stability of pINV was first confirmed by performing plasmid loss assays in vitro. Single pINV+ colonies of CVD 1233-SP were grown in broth overnight at 37 °C, and bacteria were then plated on CR agar to quantify emergence of pINV- colonies. After approximately 25 generations of growth, no pINV- colonies were detected in CVD 1233-SP, while approximately 40% of pINV- bacteria emerged in the non-stabilized pINV vaccine strain, suggesting that pINV is significantly more stable in CVD 1233-SP than in CVD 1233 (Figure 1D). The invasion and intracellular replication of these strains were then tested in HT29 cells; 53G WT and its stabilized version, 53G-SP, were included as control strains in addition to CVD 1233 and CVD 1233-SP (Figure 1A–C). There was no significant difference in the number of intracellular bacteria between CVD 1233-SP and CVD 1233 at 30 min or at 4 h (Figure 1A), indicating that stabilization of pINV in CVD 1233-SP does not influence the invasion of the vaccine strain. As with CVD 1233, the number of CVD 1233-SP bacteria recovered at 4-h post infection was not different from 30 min, indicating that this strain does not replicate intracellularly (Figure 1A). Infection with the stabilized control strain 53G-SP resulted in a significantly higher recovery at 4 h compared to 53G WT (Figure 1A), although the average fold increase for this time point was not significantly different between the two strains (Figure 1B). However, as opposed to what was previously observed for 53G WT and CVD1233, significant differences were recorded in the fold increase of the intracellular populations of 53G-SP and CVD 1233-SP (Figure 1B), suggesting that stabilization of pINV allows a better comparison between the control and the vaccine strain and confirming that unlike 53G-SP, CVD 1233-SP is not able to replicate within cells.

Furthermore, no pINV- bacteria were detected among intracellular bacteria recovered from CVD 1233-SP and 53G-SP, demonstrating that pINV is stable intracellularly when tested over the course of HT29 cell infections (Figure 1C).

Taken together, these data show that CVD 1233-SP is a promising attenuated *S. sonnei* vaccine candidate. Consistent with previous work, inactivation of the *guaBA* operon significantly reduces the intracellular replication rate of CVD 1233-SP compared to wild-type strains, while stabilization of pINV results in a dramatic decrease of spontaneous plasmid loss, providing a more accurate analysis of the invasion and replication capability of this strain and supporting its use in vaccine development.

### 2.2. Use of a Human Enteroid Model to Validate CVD 1233-SP

We tested the ability of CVD 1233, CVD 1233-SP, and the control strain 53G WT to invade and replicate within differentiated enteroid monolayers derived from human colon. Human intestinal enteroids are derived from LGR5+ stem-cell-containing colonic crypts from healthy subjects that can be differentiated into enteroids composed of enterocytes, goblet cells, and enteroendocrine cells (Figure 2 and Appendix A). This system recapitulates physiological characteristics of the intestine and represents a highly relevant human model for the evaluation of *Shigella* vaccine strains [19,24,25,26,27,28,29]. Previous enteroid infection studies performed using *S. flexneri* have shown that *Shigella* preferentially invades enteroid cells via the basolateral surface [21,22]. Therefore, differentiated enteroid monolayers were infected on the basolateral surface for 90 min and then treated with gentamicin to remove extracellular bacteria and incubated for 30 min or 4 h. Intracellular bacteria were recovered from cell lysates by plating on media containing CR, and total, pINV+, and pINV- CFU were quantified (Figure 3).

At 30 min post infection, similar numbers of intracellular bacteria were recovered for all strains, ranging between ~8 × 10^3^ and ~3 × 10^4^ CFU/mL (Figure 3A). In contrast, following 4 h of incubation, 53G WT displayed a ~10-fold increase in intracellular bacteria, reaching 2 × 10^5^ CFU/mL, while the numbers of CVD 1233 and CVD 1233-SP reduced ~10-fold to approximately 8 × 10^2^ CFU/mL (Figure 3A,B). Furthermore, there was no significant difference between 53G WT and CVD 1233 in the proportion of pINV- bacteria at both time points; more than 50% and 20% of total bacteria lost pINV in 53G WT and CVD 1233, respectively, when bacteria were recovered at 30 min, while after 4 h, pINV- bacteria accounted for only 7% of all bacteria (Figure 3C). In contrast, no pINV- colonies were detected following infection with CVD 1233-SP (Figure 3C).

These results provide further evidence that the human enteroids are a viable model to test invasion and intracellular replication of *S. sonnei* vaccine candidates. Infection of differentiated enteroid monolayers with CVD 1233-SP confirms that this strain is attenuated, as it can invade cells but is unable to replicate intracellularly compared to the wild-type strain. Stabilization of pINV in the attenuated strain significantly reduced spontaneous loss of pINV during enteroid infection, demonstrating a major advancement for the construction process of *S. sonnei* vaccine candidates.

### 2.3. Immunogenicity, Protectoin, and Safety of CVD 1233-SP in Guinea Pigs

Vaccine candidates CVD 1233 and CVD 1233-SP were tested for their immunogenicity and protective capacity in the guinea pig model. Animals were immunized with two intranasal doses of each vaccine strain approximately two weeks apart. Serum and tears were collected prior to immunization and approximately two weeks following each dose to quantify IgG and IgA responses, respectively. Following a single dose of CVD 1233 or CVD 1233-SP, 100% of animals seroconverted with 4-fold rise or greater in serum IgG (in the range of 588–1361 and 2417–7583, respectively) and mucosal IgA anti-*S. sonnei* LPS over pre-immune titres (in the range of 123–1794 and 128–1104, respectively) (Figure 4). These titres were boosted to higher levels following the second dose, reaching an average of ~1057 and ~2080 of IgA and ~14,021 and ~64,819 of IgG, respectively.

Approximately two weeks following the final dose, all animals were challenged with 53G WT using the Serény test. Following intra-ocular inoculation with ~10^8^ CFU 53G WT, all control animals developed evidence of acute infection with active inflammation and keratoconjunctivitis by 24–48-h post challenge. In contrast, 1/7 and 0/4 animals immunized with CVD 1233 or CVD 1233-SP exhibited signs of inflammation and keratoconjunctivitis. A total of 6/7 and 4/4 animals immunized with CVD 1233 and CVD 1233-SP, respectively, were completely protected against any signs of infection, resulting in efficacy values of 86% and 100% (Table 1).

### 2.4. Use of CVD 1233-SP as a Live Vector for Expression of Heterologous Antigens from ETEC

The CVD has an ongoing program to develop a combined *Shigella*-ETEC vaccine that is composed of multiple live attenuated *Shigella* strains each expressing important antigens from ETEC [30]. Based on the promising results obtained for CVD 1233-SP, this strain was used as a live vector to express the fimbrial antigens CS2 and CS3 from ETEC as a component of a multivalent *Shigella*-ETEC vaccine, producing CVD 1233-SP::CS2-CS3. The operons encoding CS2 and CS3 were engineered for high-level constitutive expression by the mLpp promoter [31] and inserted in the chromosome at a permissive site downstream of the *glmS* gene using the Tn7 system developed by McKenzie and Craig [32]. Attenuation of CVD 1233-SP::CS2-CS3 was first tested by quantifying intracellular replication in human enteroid monolayers, as described above, and compared with the control strain 53G-SP, a *S. sonnei* 53G strain with stabilized pINV (Figure 5). Similar to CVD 1233-SP, CVD 1233-SP::CS2-CS3 did not replicate intracellularly, showing a significant ~2-fold reduction in bacterial recovery following 4 h of gentamicin treatment compared to the control strain 53G-SP (Figure 5A,B).

Immunogenicity of this strain was then tested in guinea pigs. Groups of animals were immunized with two doses of CVD 1233-SP::CS2-CS3 spaced approximately two weeks apart. Serum and tears were collected for quantification of anti-*S. sonnei* LPS, CS2, and CS3-specific serum IgG and mucosal IgA responses (Figure 6). Following a single dose of CVD 1233-SP::CS2-CS3, 100% animals seroconverted with 4-fold or higher anti-*S. sonnei* LPS IgG (in the range of 69–214) and IgA titres (in the range of 21–154); titres were boosted to higher levels following the second dose, reaching, respectively, an average of ~1986 and ~245 (Figure 6A,B). Similarly, all vaccinated animals seroconverted with 4-fold or greater serum IgG responses to CS2 and CS3 following a single dose, and all titres were boosted to higher levels following a second immunization (Figure 6C,E). After two doses, all animals responded with mucosal IgA response to CS2 and CS3 (Figure 6D,F).

There is no good small-animal model to evaluate the protective efficacy against ETEC. We tested the functional capacity of serum from immunized animals to inhibit adherence of CS2- or CS3-expressing ETEC to HT29 cells [33]. Individual animal serum samples exhibited 92–99% inhibition of CS2-ETEC and 60–91% inhibition of CS3-ETEC adherence (Figure 7), which were similar to adherence-inhibition levels exhibited when bacteria were pre-incubated with commercial anti-CS2 and -CS3 antibodies (~93% and ~79%, respectively). Pre-immune serum had no inhibitory activity against ETEC attachment.

To test the efficacy of the immunization, animals previously immunized with CVD 1233-SP::CS2-CS3 and unvaccinated controls were challenged with 53G WT using the Serény test approximately two weeks following the final dose. Following intra-ocular inoculation with ~10^7^ CFU WT 53G, all (3/3) control animals succumbed to infection and exhibited inflammation and keratoconjunctivitis by 24–48-h post challenge. One of five animals immunized with CVD 1233-SP::CS2-CS3 developed infection, with 4/5 animals being completely protected against any signs of infection, resulting in an efficacy of 80% (Table 1). These data confirm that the expression of heterologous antigens in the attenuated vaccine strain CVD 1233-SP did not eliminate the ability of the live vector to induce protective responses against *Shigella*.

## 3. Discussion

In the last ten years, enteroids have emerged as a promising and revolutionary model to study enteropathogens. In contrast to human cell lines, they provide several of the characteristic features of the human intestinal epithelium, including multiple cell types, cellular composition, apical/basal cell polarization, segmental specificity, mucus secretion, and production of antimicrobial peptides [34]. The specificity of some pathogens to human hosts, including *Shigella*, often limits the use of animal models, leaving enteroids as the most physiological relevant model system to test the pathogenesis of this class of bacteria [34].

Enteroids have been especially useful for studying host-pathogen interactions with many enteric pathogens, including *Salmonella*, pathovars of *E. coli* (e.g., ETEC, EPEC, EAEC, and EHEC), and *Shigella* [34]. In particular, studies have shown that enteroids alone or in co-culture with neutrophils can be used to model *Shigella* pathogenesis and study different aspects of the early stage infection, such as epithelial cell invasion and response [21,22,35,36]. More recently, researchers have examined the use of human enteroids as preclinical models in drug and vaccine development and for studying new therapeutic approaches [37,38,39].

In this study, we employed human intestinal enteroids, for the first time, as a model system to evaluate live attenuated *S. sonnei* vaccine candidates. We used intestinal monolayers derived from LGR5+ stem-cell-containing colonic crypts from healthy subjects that were differentiated into enteroids composed of enterocytes, Paneth cells, goblet cells, and enteroendocrine cells. This system accurately reproduces the physiology of the human intestine, offering a highly relevant model for the preclinical evaluation of *Shigella* vaccine strains [19,24,25,26,27,28,29]. We previously demonstrated that a *S. sonnei* live attenuated vaccine can be produced by inactivating the de-novo pathway for biosynthesis of guanine nucleotides, which prevents intracellular replication and spread, and by eliminating expression of the ShET2 toxin [17]. We enhanced the stability of pINV in this strain by introducing mutations in one of the plasmid maintenance systems, rendering pINV more stable and resulting in CVD 1233-SP (Figure 1). Infection of differentiated enteroid monolayers was informative, as it demonstrated that this strain maintains its ability to invade cells but cannot replicate intracellularly, the critical attenuating feature of this vaccine strain (Figure 2). Stabilization of pINV reduced spontaneous loss of pINV during both in-vitro growth and infection of enteroids (Figure 1 and Figure 2), representing a significant improvement for eventual manufacture *S. sonnei* vaccine strains and potentially for challenge studies. Historically, construction of a stable *S. sonnei* vaccine strain has been problematic due to the spontaneous loss of pINV, often requiring multiple cultures to select for the most stable isolates [40,41,42,43]. The utility of this strain was further explored for the development of a combined multivalent *Shigella*-ETEC vaccine; CVD 1233-SP was used as a live vector to express the fimbrial antigens CS2 and CS3 from ETEC, producing CVD 1233-SP::CS2-CS3. Human enteroid monolayers were successfully used to confirm the attenuation of this strain, as demonstrated by the lack of intracellular replication (Figure 4). Future studies will include additional readouts from enteroid infections, including cytokine responses and differential gene expression to provide further evidence for the attenuation of this vaccine strain. Immunogenicity and protection tests of both CVD 1233-SP and CVD 1233-SP::CS2-CS3 in guinea pigs showed the induction of strong serum and mucosal antibody titres against *S. sonnei* LPS by both strains and also against CS2 and CS3 when using the multivalent strain (Figure 3 andFigure 5). Furthermore, animals immunized with either strain were protected against challenge with wild-type *S. sonnei* 53G (Table 1), and antibodies produced following immunization with CVD 1233-SP::CS2-CS3 efficiently inhibited the adherence of CS2- and CS3-expressing ETEC to HT29 epithelial cells (Figure 6). Taken together, our promising results support the advancement of these strains to further preclinical development and subsequent clinical trials.

## 4. Materials and Methods

### 4.1. Bacterial Strains and Growth Conditions

Bacterial strains used for this study are shown in the Table 2. *Shigella* strains were grown on tryptic soy broth or on tryptic soy media containing agar (TSA) (BD Difco, Franklin Lakes, NJ, USA). CVD 1233, CVD 1233-SP, and CVD 1233-SP::CS2-CS3 were grown in the presence of guanine 0.005% (*w*/*v*) (Sigma-Aldrich, St. Louis, MO, USA). Congo red (Sigma, St. Louis, MO, USA) was added to TSA media at a final concentration of 0.01% (*w*/*v*) to make CR-TSA plates. *Escherichia coli* strains were grown on lysogeny broth (LB, Invitrogen) or solid media containing 1.5% (*w*/*v*) agar (Oxoid). Antibiotics were used, when necessary, at the following concentrations: chloramphenicol, 20 μg/mL; carbenicillin, 100 μg/mL.

### 4.2. Construction of Strains

Restriction enzymes, ligases, and polymerases used for mutagenesis and PCR were purchased from New England Biolabs (Beverly, MA, USA) or Roche (Basel, Switzerland) and used according to the manufacturers’ instructions.

The details for the construction of CVD 1233 can be found in Barry et al. [17]. In CVD 1233-SP::CS2-CS3, the operons encoding CS2 and CS3 under constitutive expression by the mLpp promoter were inserted in the chromosome at a permissive site downstream of the *glmS* gene using the Tn7 system developed by McKenzie and Craig [31,32].

Integration was facilitated with plasmid pGRG-P_mLpp_-CS2-CS3. This plasmid contains the mLpp promoter, which drives high-level constitutive expression of downstream genes [31], and the operons encoding CS2 and CS3 were inserted downstream of P_mLpp_ [44]. The construct was first confirmed by PCR, restriction digestion, and DNA sequencing and then electroporated into CVD 1233-SP; single colonies were selected using procedures previously described [32]. Individual colonies were screened by PCR to detect integration of mLpp-CS2-CS3 into the CVD 1233-SP chromosome. The sequence and location of CS2 and CS3 in the chromosome of CVD 1233-SP::CS2-CS3mlpp-CS2_CS3 was confirmed by PCR and DNA sequencing.

The DNA construct for the mutagenesis of 53G-SP, CVD1233-SP, and CVD1233-SP::CS2-CS3 was generated using the primers listed in Table 3 and ligated into pUC19 using NEBuilder HIFi master mix (New England Biolabs, NEB, Ipswich, MA, USA).

The resulting plasmid was transformed into *E. coli* DH5α, and the linear DNA construct was amplified by PCR using plasmids as the template. *Shigella* strains were made electrocompetent by growing cells to a mid-log OD_600_ value of 0.5–0.8 at 37 °C and resuspending them in 1/75 the original culture volume in sterile, cold 10% glycerol following washes with same solution. Linear DNA construct was electroporated using the following conditions: 1.75 kV, 600 Ω, and 25 μF. Mutations were introduced into pINV of 53G-SP, CVD1233-SP, and CVD1233-SP::CS2-CS3 using λ-recombination [46,47]. Mutagenesis was confirmed by PCR analysis using primers binding outside of the sequences that mediated genetic recombination.

### 4.3. Congo Red Loss Assay

*S. sonnei* was grown on CR-TSA plates overnight at 37 °C to obtain single colonies. Three pINV+ colonies from each strain were re-suspended in 5mL TSB and incubated at 37 °C, 180 r.p.m., for 16 h. Samples were diluted in Dulbecco’s phosphate-buffered saline (DPBS) and plated onto CR-TSA, normalising the plating volume to the OD_600_ of the culture, and incubated overnight at 37 °C before pINV+ and pINV- colonies were counted.

### 4.4. HT29 Cell Cultivation, Invasion Assays, and Adherence Inhibition Assays

Human HT-29 (ATCC HTB-38) monolayers were cultured in DMEM (Corning, New York, NY, USA) supplemented with 10% Fetalplex (Gemini, New York, NY, USA) and 2% HEPES (Quality Biological) in 150 cm^2^ flasks (Corning, New York, NY, USA). The cells were incubated in 5% CO_2_ at 37 °C and passaged once a week. For the invasion assay, cells were seeded at a density of 5 × 10^5^ cells per well in a 24-well plate and incubated overnight. The bacterial inoculum was prepared by resuspending *S. sonnei* pINV+ colonies from a CR-TSA plate in DPBS, washing, and then suspending in DMEM adjusted for bacterial concentration of a multiplicity of infection (MOI) of 100. Therefore, a 1 mL of 5 × 10^7^ CFU/mL bacteria was added to the HT-29 cells in triplicate wells. The plates were centrifuged 5 min at 3000× *g* at 37 °C and then incubated at 37 °C with 5% CO_2_ for 90 min. The cells were then washed twice with DPBS and incubated with DMEM containing 50 μg/mL gentamicin at 37 °C with 5% CO_2_ for 30 min and 4 h. At each time point, three replicates of each sample were washed with DPBS twice to remove gentamicin and lysed with 1 mL/well of 1% Triton X-100. Serial dilutions were plated on CR-TSA to quantify total, pINV+, and pINV- intracellular CFUs of *S. sonnei*.

The ETEC adherence inhibition assay was performed as described by Poole et al. [33]. HT-29 cells were seeded at a density of 4 × 10^5^ cells/mL in a 48-well flat-bottom plate (CellTreat). ETEC strain 202,326 (expressing CS2) or strain E9034A (expressing CS3) were grown overnight at 37 °C on CFA agar (2% agar, 1% Casamino acids, 0.15% Yeast Extract, 0.005% MgSO_4_, 0.0005% MnCl_2_). Bacteria were re-suspended to a concentration of 2 × 10^8^ CFU/mL in DMEM. An 800 µL aliquot of the bacterial suspension was mixed with 80 µL individual guinea pig serum samples (pre-immunization or after second immunization) or with commercial rabbit anti-CS2 or CS3 antibodies (Bethyl and Rockland, respectively) as positive control and incubated at 37 °C for 1 h to promote binding of serum antibodies to ETEC. A 250 µL aliquot of the antibody-bacteria mixture was then used to infect HT29 cells in triplicate. The infections were carried out at 37 °C with 5% CO_2_ for 90 min. Each well was then washed with 500 µL of DPBS and allowed to shake at 544 r.p.m. for 5 min. The washing process was repeated for a total of three washes. After washing, cells were lysed for 10 min at room temperature using 500 µL of 1% Triton X-100 and serially diluted to quantify adhered bacteria. The following formula was used to calculate % of binding inhibition: 100% − [(n° bacteria^sample^/n° bacteria^PBS^) × 100%] where bacteria^PBS^ are bacteria quantified in infections performed using pre-incubated bacteria with pre-immunization sera.

### 4.5. Enteroid Cultivation and Invasion Assays

Human enteroid cultures were established from biopsies obtained after endoscopic or surgical procedures utilizing the methods developed by the laboratory of Dr. Hans Clevers [19]. De-identified biopsy tissue was obtained from healthy subjects who provided informed consent at Johns Hopkins University, and all methods were carried out in accordance with approved guidelines and regulations. All experimental protocols were approved by the Johns Hopkins University Institutional Review Board (IRB# NA_00038329). Enteroid monolayers were cultivated following the procedures described by Ranganathan et al. [21].

For the invasion assays, each transwell containing monolayers was inverted onto a sterile petri dish after aspirating the medium from the inner chamber of the transwell. The bacterial inoculum was prepared by resuspending pINV+ colonies from a CR-TSA plate in DPBS, washed, and then suspended in DMEM adjusted for bacterial concentration of a MOI of 100. The inoculum of 50 µL containing ~4.5 × 10^7^ bacteria was carefully added on the basolateral side in triplicate transwells, and cells were incubated for 2 h at 37 °C + 5% CO_2_. Following incubation, the inoculum was aspirated, and each transwell was transferred back to a 24-well plate. The cells were then washed twice with DPBS and incubated with DMEM containing 50 μg/mL gentamicin at 37 °C with 5% CO_2_ for 30 min and 4 h. After each timepoint, the transepithelial electrical resistance (TEER) was measured, the medium was aspirated, and cells were washed with DPBS twice to remove gentamicin and lysed with 1 mL/well of 1% Triton X-100 by gentle scraping to disrupt the monolayer. Serial dilutions were plated on CR-TSA to quantify total, pINV+, and pINV- intracellular CFUs of *S. sonnei*.

### 4.6. Immunizations, Sample Collection and Challenge

Overnight cultures of the immunizing strains were harvested from TSA plates with PBS to a concentration of 10^10^ CFU/mL. After sedation with ketamine/xylazine, randomized, female Hartley guinea pigs (6–8 weeks old) were immunized intranasally with 100 µL of the bacterial suspension on day 0. An identical booster dose was administered ~14 days later. Tears were collected following instillation of 50 µL of sterile PBS to the eye surface, and fluid was collected in 50 µL capillary tubes. Blood was obtained from anaesthetized guinea pigs via the anterior vena cava. Tears and blood were collected on days prior to immunization and approximately two weeks following each dose. Approximately two weeks following the second immunization, animals were challenged with a 10µL aliquot containing 8 × 10^7^–1 × 10^8^ CFU of 53G WT by intraocular administration in the Serény test. Animals were scored for signs of inflammation for 3 days as follows: 0 = normal eye indistinguishable from contralateral uninoculated eye; 1 = lacrimation or eyelid edema; 2 = 1 plus mild conjunctival hyperemia; 3 = 2 plus slight exudates; and 4 = full blown purulent keratoconjunctivitis. All animal procedures were approved by the University of Maryland, Baltimore Institutional Animal care and Use Committee, IACUC #0319002.

### 4.7. ELISA

Antigens used in enzyme-linked immunosorbent assay (ELISA) included hot water phenol-extracted *S. sonnei* LPS from WT strain 53G and purified CS2 and CS3 obtained from bei Resources (Manassas Va). Specific IgG and IgA titres were determined for serum and tears. Briefly, 96-well Immulon 2 HB plates (Thermo Milford, MA, USA) were coated with 100 µL of the antigen and incubated at 37 °C for 3 h. The plates were then washed four times with PBS containing 0.05% Tween 20 (PBST). The plates were blocked with 200 µL/well 10% milk powder (Nestle USA, Inc., Glendale, CA, USA) in PBS overnight at 4 °C. After each incubation, plates were washed four times with PBST. To determine the endpoint titres, threefold dilutions of sera or tears in 10% milk powder were added to the plates and incubated for 2 h at 37 °C. The IgG ELISA plates were then incubated with 100 μL of 1:3000 peroxidase-labeled goat anti-guinea pig IgG (Kirkegaard & Perry Laboratories, Gaitherburg, MD, USA) for 1 h at 37 °C. The plates were then incubated with substrate for 15 min at room temperature (TMB Microwell Peroxidase KPL Kirkegard & Perry Laboratories, Inc., Gaithersburg, MD, USA). Following tears incubation, IgA plates were incubated with 100 μL of 1:3000 Sheep anti guinea Pig IgA (ICL, Portland, OR, USA) for 1 h at 37 °C, then washed and incubated with 100 μL of 1:3000 peroxidase-labeled Rabbit anti Sheep IgG (ICL, Portland, OR, USA) for 1 h at 37 °C. The plates were then washed and incubated with TMB substrate for 15 min at room temperature. The reaction was stopped by the addition of 100 µL H_2_PO_4_, and the OD_450nm_ was determined in an ELISA micro plate reader (VersaMax Molecular Devices, San Jose, CA, USA). Sera and tears were run in duplicate. Linear regression curves were plotted for each sample, and titres were calculated (through equation parameters) as the inverse of the serum dilution that produces an OD_450nm_ of 0.2 above the blank.

### 4.8. Enteroid Microscopy

Enteroid monolayers (sample 109C) were prepared for staining as follows. For enterocyte and enteroendocrine staining, transwells were washed with PBS to remove loose cells and fixed with 4% paraformaldehyde (PFA) in PBS for 1 h at room temperature. PFA was removed, and transwells were stored in PBS at 4 °C until stained. Transwells were stained with phalloidin for F-actin (Invitrogen A22287) for enterocytes, chromagranin A (Developmental Studies Hybridoma Bank, University of Iowa, CPTC-CHGA-2) for enteroendocrine cells, and Hoechst 33,342 (Invitrogen H3570) for nuclei. For goblet cells and enterocyte staining, transwells were washed with PBS to remove loose cells and fixed with a 1:1 solution of cold methanol acetic acid for 10 min. Transwells were washed with PBS to rehydrate and stored in PBS at 4 °C until stained. Transwells were stained with MUC2 (Invitrogen MA5-12345) for goblet cells, EpCAM (Abcam ab223582) for lateral membranes, wheat germ agglutinin (Invitrogen W32466) for brush border, and Hoechst 33,342 (Invitrogen H3570) for nuclei.

Bacteria were labelled with carbxyfluorescein succinimidlester (CFSE) by rotating 5 mL of bacteria with 10 µM CFSE for 30 min @ 37 °C. Bacteria were centrifuged, washed with PBS 3 times, and resuspended in DMEM at a concentration of 1 × 10^9^ bacteria/mL. Stained monolayers were visualized with an Olympus FV3000 scanning laser confocal microscope using Olympus FV-31 software.

## Figures and Tables

**Figure 1 pathogens-10-01079-f001:**
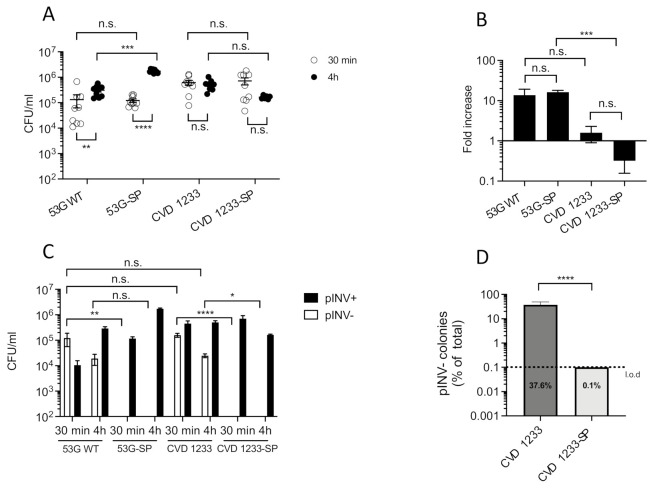
Loss of pINV is significantly reduced in CVD 1233-SP. (**A**–**C**) Intracellular replication and plasmid stability of *S. sonnei* 53G wild-type (53G WT), 53G without pINV (53G pINV-), CVD 1233, and CVD 1233 with a stabilized pINV (CVD 1233-SP) were tested in HT29 cells, and bacteria were recovered from nine independent infections performed in groups of three experiments, following 30-min and 4-h gentamicin treatment. (**A**) Number of total colonies recovered, shown as CFU/mL. (**B**) Fold increase of bacterial recovery following 4-h vs. 30-min gentamicin treatment. (**C**) Number of colonies harbouring or lacking pINV (pINV+, pINV-, respectively), shown as CFU/mL. Data are shown as mean ± S.E.M. *, *p* ≤ 0.05; **, *p* ≤ 0.01; ***, *p* ≤ 0.001; ****, *p* ≤ 0.0001; n.s., not significant; one-way ANOVA with parametric Tukey’s (**A**) or nonparametric Dunn’s (**B**,**C**) multiple comparison test. (**D**) Number of pINV- colonies emerging from CVD1233 and CVD1233-SP after growth at 37 °C for ∼25 generations in vitro, shown as a percentage of total colonies. Data are shown as mean ± S.E.M. (*n* = 9 biological replicates); ****, *p* ≤ 0.0001; *t*-test. l.o.d., limit of detection.

**Figure 2 pathogens-10-01079-f002:**
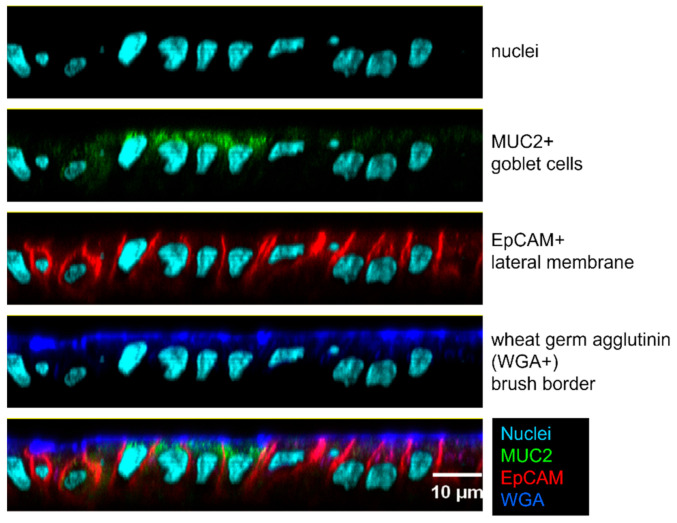
Human enteroid monolayers include multiple cell types. The enteroid monolayer was fixed with methanol and stained with Hoechst for nuclei (light blue), antibodies against MUC2 for goblet cells (green), epithelial cell adhesion molecule (EpCAM) for lateral membranes (red), and wheat germ agglutinin (WGA) for brush border (blue).

**Figure 3 pathogens-10-01079-f003:**
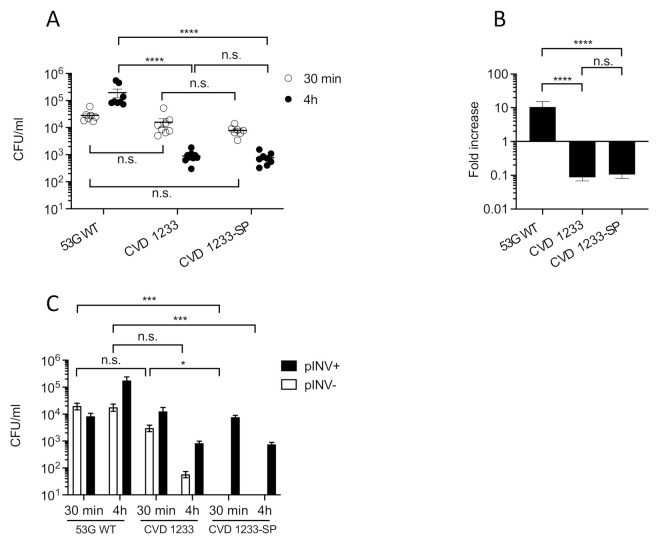
A human enteroid model to test intracellular replication and plasmid stability of CVD 1233-SP. Differentiated human enteroid monolayers were infected basolaterally with wild-type *S. sonnei* 53G (53G WT), CVD 1233, and its derivative strain with a stabilized pINV (CVD 1233-SP). Bacteria were recovered from nine independent infections performed in groups of three independent experiments, following 30-min and 4-h gentamicin treatment. (**A**) Number of total colonies recovered, shown as CFU/mL. Solid line, mean ± S.E.M. ****, *p* ≤ 0.0001; n.s., not significant; one-way ANOVA with nonparametric Dunn’s multiple comparison test. (**B**) Fold increase of bacterial recovery following 4-h vs. 30-min gentamicin treatment. Data are shown as mean ± S.E.M. ****, *p* ≤ 0.0001; n.s., not significant; one-way ANOVA with parametric Tukey’s multiple comparison test. (**C**) Number of colonies harbouring or lacking pINV (pINV+, pINV-, respectively), shown as CFU/mL. Data are shown as mean ± S.E.M. ***, *p* ≤ 0.001; *, *p* ≤ 0.05; n.s., not significant; one-way ANOVA with nonparametric Dunn’s multiple comparison test.

**Figure 4 pathogens-10-01079-f004:**
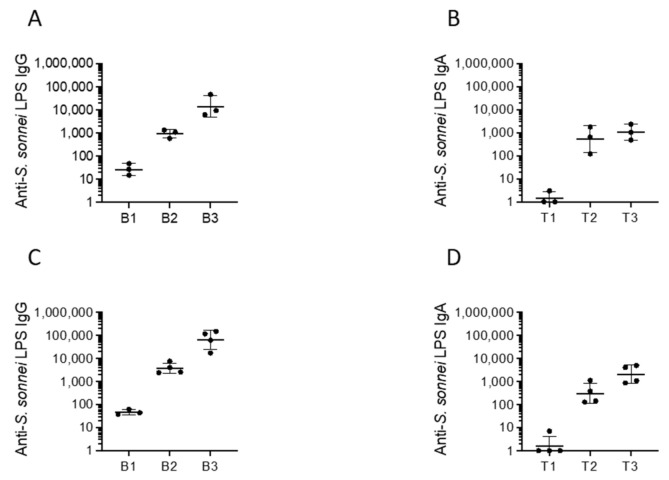
Serum and mucosal antibody responses to CVD 1233 and CVD 1233-SP. Antibody titres are shown as geometric mean titres of a group of three and four animals prior to immunization (B1 and T1) and following dose 1 (B2 and T2) and dose 2 (B3 and T3) of CVD 1233 (**A**,**B**) and CVD 1233-SP (**C**,**D**).

**Figure 5 pathogens-10-01079-f005:**
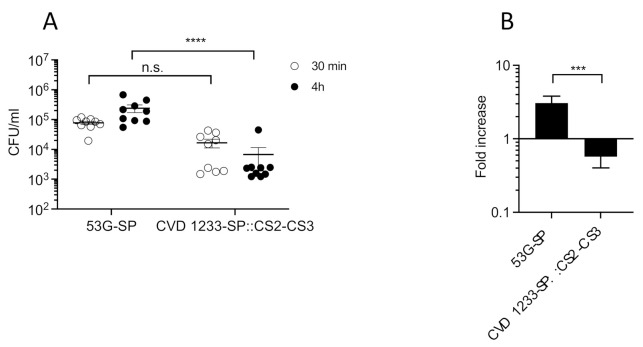
CVD 1233-SP::CS2-CS3 does not replicate in human enteroids. Differentiated human enteroid monolayers were infected basolaterally with *S. sonnei* 53G and CVD 1233::CS2-CS3, both harbouring a stabilized pINV (53G-SP and CVD 1233-SP::CS2-CS3). Bacteria were recovered from nine independent infections performed in groups of three independent experiments, following 30-min and 4-h gentamicin treatment. (**A**) Number of total colonies recovered, shown as CFU/mL. Solid line, mean ± S.E.M. ****, *p* ≤ 0.0001; n.s., not significant; one-way ANOVA with nonparametric Dunn’s multiple comparison test. (**B**) Fold increase of bacterial recovery following 4-h vs. 30-min gentamicin treatment. Data are shown as mean ± S.E.M. ***, *p* ≤ 0.001; nonparametric Mann–Whitney *t*-test.

**Figure 6 pathogens-10-01079-f006:**
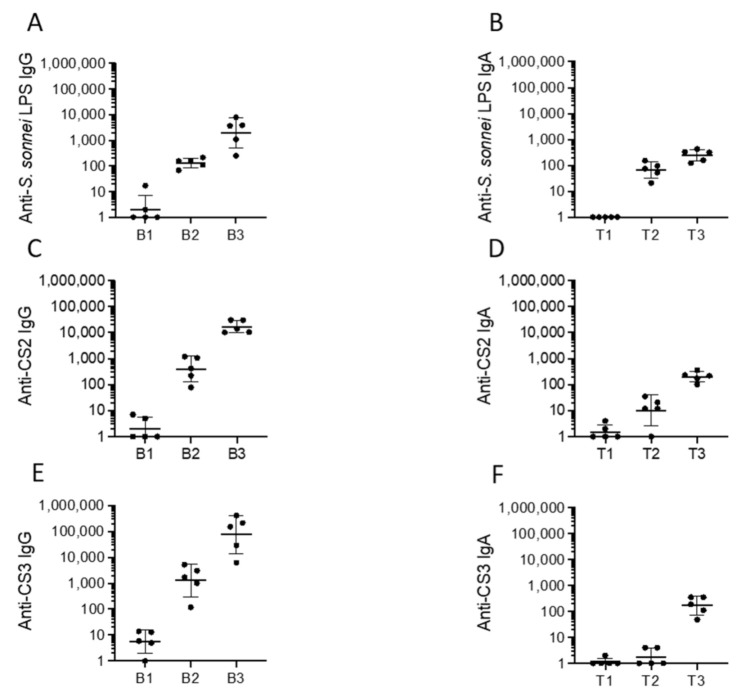
Serum and mucosal antibody responses to CVD 1233-SP::CS2-CS3. Antibody titres are shown as geometric means of titres with SD of a group of five animals prior to immunization (B1 and T1) and following dose 1 (B2 and T2) and dose 2 (B3 and T3) of CVD 1233-SP::CS2-CS3. Antibody titres are shown for *S. sonnei* LPS (**A**,**B**), CS2 (**C**,**D**), and CS3 (**E**,**F**).

**Figure 7 pathogens-10-01079-f007:**
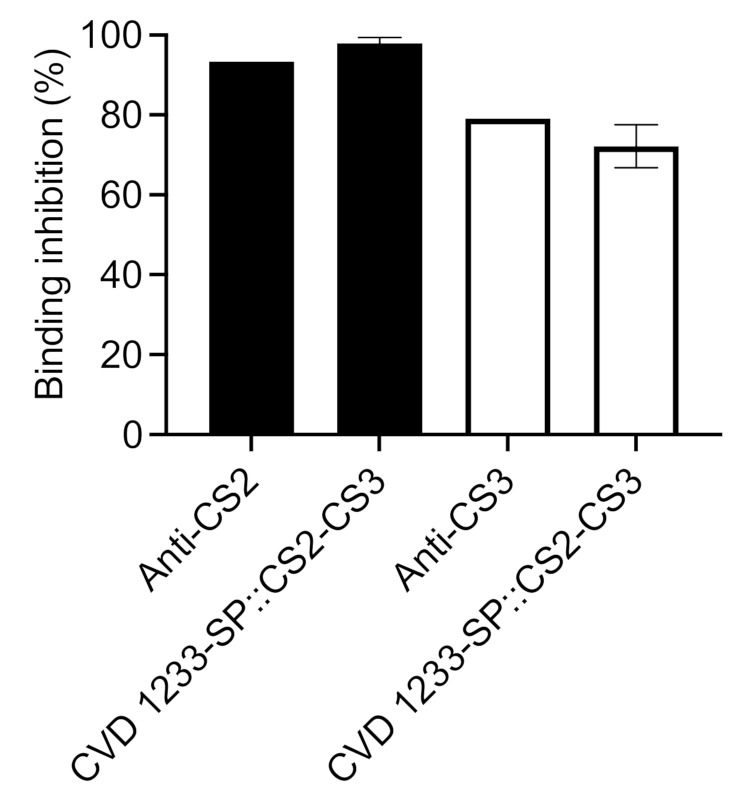
Serum-mediated inhibition of adherence of CS2- and CS3-expressing ETEC. Sera from five guinea pigs previously immunized with CVD 1233-SP::CS2-CS3 were tested to quantify binding inhibition of adherence of ETEC strains expressing CS2 (strain 202326, black bars) or CS3 (strain E9034A, white bars). As a control, binding inhibition was also tested using bacteria pre-incubated with commercial anti-CS2 and anti-CS3 antibodies. Bars indicated as mean ± S.E.M.

**Table 1 pathogens-10-01079-t001:** Protection of animals immunized with CVD 1233, CVD 1233-SP, and CVD 1233::CS2-CS3-SP following challenge with *S. sonnei* strain 53G WT.

Treatment Group	Attack Rate	Efficacy
CVD 1233 ^§^	1/7	86%
CVD 1233-SP	0/4	100%
Unvaccinated controls ^§^	6/6	-
CVD 1233-SP::CS2-CS3	1/5	80%
Unvaccinated controls	3/3	-

Efficacy = ARU − ARV/ARU × 100, ^§^ Data combined from 2 separate studies.

**Table 2 pathogens-10-01079-t002:** List of strains used in this study.

Strain	Description	Reference
53G WT	Wild-type *S. sonnei* 53G	[45]
53G-SP	53G WT with stabilized pINV	This work
CVD 1233	Δ*guaBA*, Δ*sen*, *S. sonnei* vaccine derived from 53G WT	[17]
CVD 1233-SP	CVD 1233 with stabilized pINV	This work
CVD 1233::CS2-CS3	CVD 1233-expressing CS2 and CS3 antigens from ETEC	This work
CVD 1233-SP::CS2-CS3	CVD 1233::CS2-CS3 with stabilized pINV	This work

**Table 3 pathogens-10-01079-t003:** List of primers used in this study to generate strains.

Primer	Sequence (5′-3′)	Generated Strains
Wu215_F	ACCGAACAACGAACTGTTGGAA	CVD 1233::CS2-CS3 and CVD 1233-SP::CS2-CS3
Wu215_R	TGCGTAGCGTTACAGTACCTGAT	
Giulia007	acggccagtgaattcgagctGTGAAGCGGGTCCGGGTG	53G-SP, CVD 1233-SP and CVD 1233-SP::CS2-CS3
GP173	ATGGAAACCACCGTATTTCTCAGCAACCGCAGC	
GP174	GCTGCGGTTGCTGAGAAATACGGTGGTTTCCAT	
JM296	AACTTCAGCATCAGAATGACTCCCTTTC	
JM298	GTCATTCTGATGCTGAAGTTTATGCTCGATACCAAC	
GM174	TGTGTAGGCTGGAGCTGCTT	
GM175	ATGGGAATTAGCCATGGTCC	
Giulia010	ccatggctaattcccatTCAGCTCCAGTCTTCAGTTC	
Giulia011	agcctacacaCCTGTTCATCAGAAATCATCTC	
GP141	CATGATTACGCCAAGCTgtaatcgtcgtgtgtgg	
GP175	gcaatggcctgtgtcacc	
GP176	atgtggctgtccattgcc	

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
