# Peer review of "Evaluation of a Live Attenuated S. sonnei Vaccine Strain in the Human Enteroid Model"

_pathogens, 2021, doi:10.3390/pathogens10091079_

Round 1

Reviewer 1 Report

Evaluation of a live attenuated S. sonnei vaccine strain 2 in the human enteroid model

Scope: The study describes the construction of a modified and plasmid-stabilzed Shigella sonnei live attenuated vaccine strain, CVD 1233-SP and its multivalent derivative, CVD 1233-SP::CS2-CS3, and, implicate their use as successful candidates for Shigella vaccine.

Both, CVD 1233-SP and CVD 1233-SP::CS2-CS3   were further evaluated in vitro in the human enteroid GI model for their ability to invade but not replicate in these cells.

The constructs were also tested for immune responses using the guinea pig experimental model for both IgA and IgG esponses.

Comments:

It is an elaborate study that has been reasonably well designed and conducted. It is also well presented.

Minor points:

The authors must correct mistakes that are typographic or wrongly expressed.

Page 1, line 19 delete ‘virulence’

Page 1, line 45 replace ‘regions transition’ with regional transition

Elaborate ‘ETEC’ when it is referred to for the first time as Enterotoxigenic Eschericia coli

Author Response

REVIEWER #1

Comments and Suggestions for Authors

Evaluation of a live attenuated S. sonnei vaccine strain 2 in the human enteroid model

Scope: The study describes the construction of a modified and plasmid-stabilzed Shigella sonnei live attenuated vaccine strain, CVD 1233-SP and its multivalent derivative, CVD 1233-SP::CS2-CS3, and, implicate their use as successful candidates for Shigella vaccine.

Both, CVD 1233-SP and CVD 1233-SP::CS2-CS3   were further evaluated in vitro in the human enteroid GI model for their ability to invade but not replicate in these cells.

The constructs were also tested for immune responses using the guinea pig experimental model for both IgA and IgG esponses.

Comments:

It is an elaborate study that has been reasonably well designed and conducted. It is also well presented.

Minor points: Author responses are in red below each comment.

The authors must correct mistakes that are typographic or wrongly expressed.

Page 1, line 19 delete ‘virulence’

This edit has been made

Page 1, line 45 replace ‘regions transition’ with regional transition

This edit has been made.

Elaborate ‘ETEC’ when it is referred to for the first time as Enterotoxigenic Eschericia coli

This edit has been made

Reviewer 2 Report

The work is interesting and well developed. Validation assays on Shigella-vaccine strains and in vivo effectiveness of vaccination in animal models are conclusive and thorough. Still, visual presentation of the human enteroid model and its interaction with bacteria is an insufficiency difficult to overlook.

Major remarks:

  • In the Results section quantitative data is overwhelming for the reader, majority of the numerical data (mainly in the brackets) should be moved to figure legends.
  • Event though the quantitative approach is rigourous with multiple experiments and the results are convincing, the credibility of the whole work is limited with no visual interpretation at all of the experiments. The eligibility of the human enteroid modell highly depends on the diverse cell-types present in the intestinal mucosa, including enterocytes, goblet cells, enteroendocrine cells and Paneth cells. All these cell populations are easily identifiable by routine histological stainings (HE), without the need for resource-intensive immunoassays. Shigella species can be visualized by simple Giemsa stainings or other methods (https://europepmc.org/article/PMC/6434139)

Minor remarks:

  • Please correct typo in abstract "stabilzed" in line 19
  • Please remove doubled "of" in line 68

Author Response

We appreciate the reviewer’s comments and have responded to individual comments with red text below each specific suggestion.

REVIEWER #2

Comments and Suggestions for Authors

The work is interesting and well developed. Validation assays on Shigella-vaccine strains and in vivo effectiveness of vaccination in animal models are conclusive and thorough. Still, visual presentation of the human enteroid model and its interaction with bacteria is an insufficiency difficult to overlook.

Major remarks:

  • In the Results section quantitative data is overwhelming for the reader, majority of the numerical data (mainly in the brackets) should be moved to figure legends.

The quantitative data has been removed from the text as directed and is contained within figures and figure legends.

  • Event though the quantitative approach is rigourous with multiple experiments and the results are convincing, the credibility of the whole work is limited with no visual interpretation at all of the experiments. The eligibility of the human enteroid modell highly depends on the diverse cell-types present in the intestinal mucosa, including enterocytes, goblet cells, enteroendocrine cells and Paneth cells. All these cell populations are easily identifiable by routine histological stainings (HE), without the need for resource-intensive immunoassays. Shigella species can be visualized by simple Giemsa stainings or other methods (https://europepmc.org/article/PMC/6434139)

Figure 2 and supplementary figures 1 and 2 have been added to the manuscript to demonstrate visualization of the multiple cell types within the enteroid monolayer. We agree this is an important addition to this manuscript. We would like to explain that we did not originally include these figures because we follow a validated protocol that results in the differentiation of cell types within the monolayer (Saxena et al, J Virol, 2015; Zachos, J Biol Chem, 2016; In, Nat Rev Gastroenterol Hepatol, 2016). Each monolayer is very valuable and as it is not easy to produce a very high number of replicates and we did not want to give up an experimental replicate. The controls have now been completed and the manuscript is strengthened by this addition.

Minor remarks:

  • Please correct typo in abstract "stabilzed" in line 19

The correction has been made

  • Please remove doubled "of" in line 68

The correction has been made

Round 2

Reviewer 2 Report

The authors has sufficiently fulfilled the requested revisions for the paper. The reviewer agrees with the authors regarding the value of monolayers and the need for their preservation, however I still thinks the manuscript benefited from the extra figures and succeded to deliver a more illustrative presentation of the study. The authors effort to improve the manuscript with microscopic pictures of the enteroid models is appreciated and well-taken.